# Genome-Wide Identification and Characterization of Wheat 14-3-3 Genes Unravels the Role of TaGRF6-A in Salt Stress Tolerance by Binding MYB Transcription Factor

**DOI:** 10.3390/ijms22041904

**Published:** 2021-02-14

**Authors:** Wenna Shao, Wang Chen, Xiaoguo Zhu, Xiaoyi Zhou, Yingying Jin, Chuang Zhan, Gensen Liu, Xi Liu, Dongfang Ma, Yongli Qiao

**Affiliations:** 1Engineering Research Center of Ecology and Agricultural Use of Wetland, Ministry of Education, Hubei Collaborative Innovation Center for Grain Industry, College of Agriculture, Yangtze University, Jingzhou 434000, China; 201871379@yangtzeu.edu.cn (W.S.); 201871384@yangtzeu.edu.cn (X.Z.); 202072787@yangtzeu.edu.cn (Y.J.); 202072788@yangtzeu.edu.cn (C.Z.); 202071677@yangtzeu.edu.cn (G.L.); 202072786@yangtzeu.edu.cn (X.L.); 2Shanghai Key Laboratory of Plant Molecular Sciences, College of Life Sciences, Shanghai Normal University, Shanghai 200234, China; zhuxiaoguo1990@126.com; 3Oil Crops Research Institute, Chinese Academy of Agricultural Sciences, Wuhan 430062, China; chenwangchw@163.com

**Keywords:** 14-3-3 gene family, *Triticum aestivum* L., bioinformatics analysis, salt tolerance, protein-protein interactions

## Abstract

14-3-3 proteins are a large multigenic family of general regulatory factors (GRF) ubiquitously found in eukaryotes and play vital roles in the regulation of plant growth, development, and response to stress stimuli. However, so far, no comprehensive investigation has been performed in the hexaploid wheat. In the present study, A total of 17 potential 14-3-3 gene family members were identified from the Chinese Spring whole-genome sequencing database. The phylogenetic comparison with six 14-3-3 families revealed that the majority of wheat *14-3-3* genes might have evolved as an independent branch and grouped into ε and non-ε group using the phylogenetic comparison. Analysis of gene structure and motif indicated that 14-3-3 protein family members have relatively conserved exon/intron arrangement and motif composition. Physical mapping showed that wheat *14-3-3* genes are mainly distributed on chromosomes 2, 3, 4, and 7. Moreover, most *14-3-3* members in wheat exhibited significantly down-regulated expression in response to alkaline stress. VIGS assay and protein-protein interaction analysis further confirmed that TaGRF6-A positively regulated slat stress tolerance by interacting with a MYB transcription factor, TaMYB64. Taken together, our findings provide fundamental information on the involvement of the wheat 14-3-3 family in salt stress and further investigating their molecular mechanism.

## 1. Introduction

General regulatory factor (GRF) proteins, also known as 14-3-3 proteins, are found among all eukaryotic organisms [1]. The 14-3-3 protein, first discovered in the bovine brain as a soluble acidic protein, acquired its name based on the fraction number in diethylaminoethyl (DEAE) cellulose chromatography and migration position on starch-gel electrophoresis [2]. Initially, 14-3-3 proteins were believed to be unique to animals. However, further research showed that these proteins are found in almost all eukaryotes [1,3,4]. The 14-3-3 proteins are a class of highly conserved regulatory proteins that can form homo- or heterodimers, and each monomer in the dimer can interact with a separate target protein. This dimeric property of 14-3-3s allows them to serve as scaffolds for bringing together different regions of a protein or two different proteins in close proximity [5,6]. Therefore, 14-3-3 proteins regulate activities of numerous target proteins via protein-protein interactions, specifically by binding to the phosphoserine/phosphothreonine residues of target proteins. Evidence suggests that 14-3-3s are the most important phosphopeptide-binding proteins that play a regulatory role in various biological processes including the regulation of cell cycle, metabolism, apoptosis, development, protein trafficking, stress response, and gene transcription [7,8,9,10,11,12,13,14,15].

In plants, the first plant 14-3-3 protein was isolated from *Arabidopsis thaliana* as a component of the G-box complex and was named GF14 [16]. Subsequently, 14-3-3 proteins were detected in many plant species and were named GF or GRF, followed by Arabic numerals [17]. With the development of new high-throughput sequencing technologies and modern bioinformatics tools, more and more plant genomes were sequenced, which dramatically accelerated the genome-wide survey of 14-3-3 proteins in plants. To date, the 14-3-3 gene family has been identified in *Arabidopsis* (13), rice (*Oryza sativa* L.) (8), tomato (*Solanum lycopersicum* L.) (12), soybean (*Glycine max* L.) (18), cotton (*Gossypium hirsutum* L.) (25), banana (*Musa acuminata* L.) (25), grape (*Vitis vinifera* L.) (11), and many other plant species [18].

Wheat (*Triticum aestivum* L.) is the most widely cultivated crop in the world and one of the primary grains consumed by humans [19]. Drought, extreme temperatures, and salinity are the major abiotic stresses that reduce wheat production throughout the growing season [20]. The 14-3-3 proteins act as key regulators of signaling networks and abiotic stress responses [21]. For example, in rice, OsGF14e interacts with OsCPK21 to promote the salt stress response [22]. Additionally, studies show that OsGF14f has a negative effect on grain development and filling, while OsGF14e negatively impacts cell death and disease resistance [23,24]. In tomato, four 14-3-3 genes (*TFT1*, *TFT4*, *TFT7*, and *TFT10*) are significantly up-regulated under salt stress [25]. Ectopic expression of the wheat *TaGF14b* gene in tobacco (*Nicotiana tabacum* L.) enhanced the tolerance of mature tobacco plants to drought and salt stresses, which are related to the abscisic acid (ABA) signaling pathway, and improved their growth and survival compared with the control [26]. Overexpression of the soybean *GsGF14o* gene in *Arabidopsis* showed that GsGF14o regulates stomata size, root hair development, and drought tolerance [27].

Recently, Guo et al. [28] analyzed wheat *TaGF14* genes based on the TGACv1 reference genome and found that five of these genes were up-regulated under drought stress, and all of the analyzed *TaGF14s* were down-regulated during heat stress. These results suggest that some of the *TaGF14* genes play a vital role in combating drought stress, and all *TaGF14s* are negatively associated with heat stress. Additionally, the study showed that TaGF14s might be involved in starch biosynthesis. However, no information is available on the structure of 14-3-3 genes and their role in response to salt tolerance in wheat.

In this study, we conducted an in-depth analysis of the wheat 14-3-3 gene family members based on the newly released Chinese Spring reference genome (IWGSC RefSeq v1.1, https://urgi.versailles.inra.fr/download/iwgsc/IWGSC_RefSeq_Annotations/v1.1/ (accessed on 13 February 2021)). We performed a comprehensive analysis of the structure, evolutionary relationship, *cis*-acting elements, Gene Ontology (GO) annotations, and expression profiles of the *14-3-3* genes, and deciphered the biological roles of these genes in salt stress tolerance using the virus-induced gene silencing (VIGS) assay and protein-protein interaction analysis. It provided an updated view of the wheat 14-3-3 family and lays the foundation for studying the molecular mechanism of *TaGRF6-A* in regulating salt tolerance in wheat.

## 2. Results

### 2.1. Genome-Wide Identification and Characterization of 14-3-3 Family Members in Wheat

A total of 32 14-3-3 protein sequences of wheat were identified using the basic local alignment search tool, BLASTp, and validated using the Hidden Markov Model (HMM) search tool, HMMER. These sequences were further confirmed using SMART and NCBI-CDD online tools, which revealed that all sequences contained conserved 14-3-3 protein domains. These protein sequences were encoded by 17 genes, including 12 genes showing alternative splicing, and splice variants with complete domains were chosen as representatives (Table 1). Multiple sequence alignment and secondary structure analysis revealed that the 14-3-3 family members contained nine typical α-helices. But the *TraesCS4A02G167100* gene lacked the first α-helix, making it the first 14-3-3 family member that does not contain nine α-helices (Appendix A). The 14-3-3 amino acid sequences were highly conserved, indicating that these proteins may perform similar functions to the 14-3-3 proteins of other plant species. The C-terminal end of 14-3-3 proteins was relatively divergent, which explains the functional diversity observed among these proteins. Genes with a 1:1:1 correspondence in all three sub-genomes (A, B, and D) of wheat are called triads [29]. We identified five triads based on the results of Ramírez-González et al. (Appendix A). Because of the peculiarity of *TraesCS4A02G167100*, this gene was named *TaGRF-like1*. The remaining 16 genes were named according to their order on various homologous chromosomes (Table 1).

A detailed description of *TaGRFs* is summarized in Table 1. The deduced wheat 14-3-3 proteins contained 244–282 amino acid residues, and their molecular weight (MW) ranged from 27.25 to 31.93 kDa. The predicted isoelectric point (pI) of these proteins ranged from 4.67 to 6.32, implying that these proteins were acidic in nature. Three-dimensional modeling clearly showed the presence of α-helices and indicated that higher structures of these proteins were very similar (Appendix A).

### 2.2. Gene Structure and Conserved Motif Analysis

Phylogenetic analysis revealed that *TaGRFs* were divided into two groups (ε and non-ε) and three corresponding members of each triad were closely clustered together (Figure 1A). To examine the structural diversity of *TaGRFs*, we manually extracted the exon-intron structure from the GFF3 annotation file (Appendix A). All *TaGRF* genes, especially those within the same phylogenetic group, shared a similar exon-intron structure (Figure 1B). The number of exons varied from 4 to 6 and the genes within each phylogenetic group exhibited nearly identical exon lengths. To understand the motif composition, a search was implemented using the MEME website, with the maximum number of motifs set at 10 (Figure 1C). Functional analysis using SMART revealed that 3 of these 10 motifs (motifs 1–3) were annotated as 14-3-3, which were included in all members (Appendix A). This result suggests that motifs 1–3 represent the main domains that determine the function of 14-3-3 proteins. At the same time, this result also indicates the functional conservation of the *14-3-3* genes during evolution. In addition, motif composition was highly similar within each phylogenetic group but was relatively divergent between groups. All members of the non-ε group contained six main motifs (1–6), whereas members of the ε group contained two unique motifs (9 and 10), which indicates the potentially different roles during plant growth and response to various stresses.

### 2.3. Chromosomal Distribution and Evolutionary Analysis of TaGRF Genes

To clarify the chromosome position of *TaGRFs*, a schematic diagram was constructed. The *TaGRF* genes were mainly distributed on homoeologous chromosomes 4A/4B/4D, and only one gene was located on each of the remaining chromosomes (Figure 2A).

To trace the evolutionary origin of wheat *TaGRFs*, we performed a synteny analysis of family members in common wheat (AABBDD) and its progenitor species *Aegilops tauschii* L. (DD), *Triticum urartu* L. (AA), and *T. dicoccoides* L. (AABB) (Figure 2B). Given that the B sub-genome donor has not been sequenced, wild emmer wheat (*T. dicoccoides* L.) was used as the source of the B sub-genome. The results showed that *TaGRFs* had seven, twelve, and five orthologs in *T. urartu*, *T. dicoccoides*, and *Ae. tauschii*, respectively (Appendix A). Interestingly, the origin of *TaGRF* genes was traceable. Our results indicated that the *TaGRF* genes in common wheat were entirely derived from the sub-genome donors and have the same number on the corresponding chromosomes. Wheat chromosomes 4A, 4B, and 4D harbored three, three, and two *TaGRF* genes, respectively. The same number of *GRF* homologs were detected on chromosomes 4A and 4B of *T. dicoccoides* and on chromosome 4D of *Ae. tauschii*. However, the corresponding gene was not detected on chromosome 2A of *T. urartu*. We speculated that the genes on chromosome 2A of common wheat evolved from *T. dicoccoides* or the corresponding gene on *T. urartu* used to exist but was later lost. This implied that members of the *TaGRF* gene family did not experience large fluctuations. The only polyploidization doubles the number of genes twice. In addition, the non-synonymous-to-synonymous substitution ratio (Ka/Ks) of *TaGRF* gene duplicates was <1 (Figure 2C, Appendix A), indicating that these genes are under negative selection pressure to maintain the protein sequence.

To understand the evolutionary relationships of wheat 14-3-3 proteins with those of other plant species, a phylogenetic tree of full-length 14-3-3 amino acid sequences from seven plant species, including *T. aestivum*, *A. thaliana*, *O. sativa*, *G. max*, *Populus trichocarpa*, *Medicago truncatula*, and *V. vinifera*, was constructed using the maximum likelihood (ML) method (Figure 2D, Appendix A). Consistent with the previous classification [30], all 14-3-3 proteins were divided into two main evolutionary branches, namely, ε and non-ε. Only *TaGRF-like1* and *TaGRF5-B* were grouped into the ε group, and the remaining *TaGRFs* clustered in the non-ε group. The proteins from wheat, *Arabidopsis*, and rice were closely related, suggesting that these proteins may perform similar functions, which provides clues for functional analysis.

### 2.4. Cis-Element Analysis and Functional Annotation of TaGRF Genes

To examine *cis*-acting elements in *TaGRF* promoter sequences, we searched 1.5 kb sequence upstream of the start codon (ATG) of all *TaGRF* genes. A total of 1789 *cis*-acting elements were identified (Appendix A). These included elements involved in biotic and abiotic stress responses, plant growth and development, and phytohormone response processes. The number of elements involved in the first two processes was significantly more (Figure 3A,B). Motifs such as TGACG and CGTCA (involved in responding to methyl jasmonate [MeJA]) and ABA-responsive element (ABRE; involved in the response to ABA) were commonly found in *TaGRF* gene promoters, suggesting that these genes are involved in hormone regulation. In addition, stress-related *cis*-elements, such as Myb, STRE, Sp1, G-box, Myc, as-1, LTR, and GC-motif, were also identified, implying that *TaGRFs* are important players in response to adverse conditions.

To further predict the function of *TaGRFs*, we performed GO annotation analysis. The results showed that *TaGRFs* were divided into three categories (molecular function, cell composition, and biological process) (Appendix A). In the molecular function category, GO terms such as “ATP binding” and “protein domain specific binding” were highly enriched, indicating that TaGRF proteins may bind to other proteins to perform various cellular functions. In the cell component category, GO terms such as nucleus, cytosol, plasma membrane, and organelles, such as mitochondrion, were highly enriched, suggesting that *TaGRFs* perform a wide range of functions. In the biological process category, *TaGRFs* play a prominent role in the response to cadmium ions. Both *TaGRF5-B* and *TaGRF-like1* were annotated only under the GO term “protein domain specific binding”, suggesting that these proteins play a limited role in wheat.

### 2.5. Expression Profiling of TaGRF Genes

The results of *cis*-element and GO annotation analyses suggested that *TaGRFs* are potentially involved in abiotic stress responses. To better understand the function of *TaGRFs*, we examined their expression patterns using RNA-seq data. The expression profiles of 16 *TaGRFs* in different wheat varieties under salt stresses are shown in Figure 4A. In general, the expression of *TaGRFs* decreased after the salt treatment. Furthermore, the expression levels of *TaGRF4-A*, *TaGRF4-B*, and *TaGRF5-B* were extremely low. In the wheat variety Kharchia Local, *TaGRFs* showed different expression profiles in different tissues; for example, *TaGRF1-B*, *TaGRF1-D*, and *TaGRF2-A* showed low expression in roots, while *TaGRF6-A*, *TaGRF6-B*, and *TaGRF6-D* showed high expression in leaves, which indicated that these genes may perform a function mainly in leaves. Compared with Kharchia Local, QM6 and Chinese Spring showed greater differences in gene expression after the salt treatment, suggesting that genotype is an important factor affecting the differences in abiotic stress responses.

To verify the reliability of the transcriptome data, we analyzed the expression of two genes (*TaGRF4-D* and *TaGRF6-A*) in 1-week-old salt-treated wheat seedlings by quantitative real-time PCR. Both the *TaGRFs* responded to salt stress, but with different rates and intensities (Figure 4B). Compared with the control, the expression of *TaGRF4-D* did not change significantly, while the expression of *TaGRF6-A* was significantly up-regulated at 12 h post salt treatment, and then decreased sharply at 24 h post salt treatment. This trend suggests that *TaGRF6-A* can be induced by salt stress at a certain stage, and may play a role during this period.

### 2.6. TaGRF6-A Positively Regulates Salt Tolerance in Wheat

The VIGS assay is used to silence specific genes in plants, thus enabling rapid characterization of gene function [31]. To further understand the role of *TaGRF6-A* in salt stress, we performed a virus-induced gene silencing assay. Two fragments of the *TaGRF6-A* gene were chosen for silencing (Appendix A). At 10 days post-inoculation (dpi), wheat leaves inoculated with BSMV showed mild chlorotic mosaic symptoms while leaves inoculated with BSMV: *TaPDS* showed bleaching symptoms, indicating that the BSMV-induced gene silencing system was working (Figure 5A). When treated with 300 mM NaCl for 12 days, both fragments of *TaGRF6-A* were well silenced by qRT-PCR (Figure 5C), and at this time, the leaf curling degree of plants carrying BSMV: *TaGRF6-A-1/2* was stronger than that of the control group carrying the empty vector (Figure 5B). According to the statistical results, the number of leaf curls of plants carrying BSMV: *TaGRF6-A-1/2* was significantly more than that of the empty carrier group (Figure 5D), indicating that *TaGRF6-A* contributes to salt tolerance in wheat.

### 2.7. TaGRF6-A Interacts with TaMYB64 In Vitro and In Vivo

Since *TaGRF6-A* expressed salt tolerance, we started to look for its internal mechanism. We focused on the expression levels of stress-related genes and their potential targets (Figure 6A). It is worth noting that *TaMYB64* is significantly down-regulated, suggesting that there may be a synergistic effect to enhance abiotic stress tolerance together with *TaGRF6-A*. Previous studies in soybean and rice showed that interactions between MYB transcription factors and 14-3-3 proteins enhance plant growth and stress tolerance [32,33]. These findings, combined with our qRT-PCR results, prompt us to investigate the association between TaMYB64 and TaGRF6-A. Before that, we analyzed the subcellular localization of TaGRF6-A and TaMYB64 by generating a C-terminal fusion of gene and yellow fluorescent protein (YFP) gene. After transient expression in *N. benthamiana* leaves using *Agrobacterium*-mediated transformation, compared with the YFP empty vector, confocal microscopy showed that TaGRF6-A has a similar subcellular localization and was detected in both the nucleus and the cytoplasm (Figure 6B), and TaMYB64 was detected in the nucleus, indicating the place where it functions. When exploring the interaction of TaMYB64 and TaGRF6-A, we first performed yeast two-hybrid (Y2H) assays. Although yeast transformants expressing both TaMYB64 and TaGRF6-A could grow on high stringency media, the number of yeast cells was less (Figure 6C). To further verify the interaction between TaMYB64 and TaGRF6-A, we performed bimolecular fluorescence complementation (BiFC) and co-immunoprecipitation (Co-IP) assay. After transiently co-expressing TaGRF6-A labeled with the N-terminal half of YFP (nYFP), together with TaMYB64 labeled with the C-terminal half of YFP (cYFP) in *N. benthamiana* leaves, the YFP signal was detected in the nucleus and cytoplasm of *N. benthamiana* leaf epidermal cells. While no signal was detected in the control combinations (Figure 6D). The TaMYB64-YFP-3×FLAG fusion proteins were co-expressed with TaGRF6-A-YFP-HA or YFP-HA in *N. benthamiana*. After immunoprecipitation using the anti-HA antibody, Western blot was performed using the anti-FLAG antibody to detect TaMYB64 proteins. The TaMYB64-YFP-3×FLAG protein could be coimmunoprecipitated by TaGRF6-A-YFP-HA, but not the YFP-HA control, revealing a specific interaction between TaMYB64 and TaGRF6-A in vivo (Figure 6D). These results proved that TaGRF6-A could physically interact with TaMYB64.

### 2.8. TaGRF6-A and TaMYB64 Work Together to Cope with Salt Stress

To clarify whether TaGRF6-A interacts with TaMYB64 to respond to salt stress, we chose to silence the *TaMYB64* gene. Ten days post-inoculation (dpi), wheat leaves inoculated with BSMV showed mild symptoms of chlorosis, while leaves inoculated with BSMV: *TaPDS* showed symptoms of bleaching stripes, indicating that BSMV-induced gene silencing was effective (Figure 7A). qRT-PCR results revealed that both fragments of *TaMYB64* were successfully silenced (Figure 7C). After treated with 300 mM NaCl for 12 days, the plants carrying BSMV: *MYB64-1/2* had stronger curls and a significantly higher number of rolled leaves compared with the control group carrying the empty vector (Figure 7B,D). These results indicated that *TaMYB64* was also involved in response to salt stress.

## 3. Discussion

14-3-3 genes are ubiquitous in many species, and the study of its gene function is very necessary. Recent studies have shown that the four 14-3-3 proteins (MdGF14a, MdGF14d, MdGF14i, and MdGF14j) of apples interact with the flower integrator TFL1/FT and participate in flowering regulation [34]. Under the condition of nitrate deficiency, the interaction of MdGRF11 and MdBT2 in apples can induce the accumulation of anthocyanins and provide a new mechanism for anthocyanin biosynthesis [35]. Under sugar starvation, MYBS2 interacted with 14-3-3 protein to inhibit the expression of α-amylase, thereby to improve rice plant growth, stress resistance, and grain weight [32]. As the hub of different signal pathways, 14-3-3 protein can transmit and integrate different hormone signals in plant signal transduction [36]. 14-3-3 interacts with AtWRI1, and the biosynthesis of oil increased in transgenic tobacco leaves co-expressing these two genes [37]. 14-3-3λ and k are important salt tolerance regulators. When there is no salt stress, 14-3-3λ and k interact with SOS2 and inhibit the kinase activity of SOS2; while in the presence of salt stress, the interaction of them weakens and activates SOS2 kinase activity [38].

In the present study, we performed a genome-wide characterization and in silico analysis of the *14-3-3* gene family in wheat. A total of 17 *14-3-3* genes were identified in wheat based on the genome sequence of Chinese Spring. However, this result is inconsistent with previous studies, which identified 20 *14-3-3* genes based on the TGACv1 wheat genome assembly [28]. We found that previous results contained duplicate genes and several other genes that do not exist in our newly released genome sequencing data. For example, Traes_2DL_1D912517E and Traes_2DL_21639EB55, which were previously identified as two distinct genes, represent the same gene in the new genome. In addition, TraesCS2D02G325600 and Traes_3B_3BF024700350CFD_c1 are not identified in the present study. We speculate that this difference between studies is caused by differences in genome sequencing depth, genome analysis, and gene annotations. We also cannot rule out the possibility of wheat germplasm resource differences.

The 14-3-3 proteins are highly conserved and usually contain nine α-helices, which constitute the conserved core region of each monomer in the dimer [13,39]. In the current study, we showed that *TaGRF-like1* lacks the first α-helix, which may affect the binding of phosphorylated target proteins. while the secondary structure of TaGRF proteins was conserved except in the C-terminal region (Appendix A). Previous studies have shown that the variable C-terminus is the key to dimer formation, and it can interact with different ligands to show the target specificity of 14-3-3 protein [40,41].

Phylogenetic analysis of 89 14-3-3 proteins belonging to wheat and six other plant species showed that these proteins were clustered into two major groups (ε and non-ε), which is consistent with previous reports [18,34]. The result indicates that the formation of these two isoforms is a basic and ancient difference (Figure 2D). As *TaGRF6-A* was adjacent to *OsGF14d*, which is expressed under salt, heat, and cold stress [42], the function of TaGRF6-A may be similar to that of *OsGF14d*. According to previous studies, the ε group genes usually contain a greater number of exons and motifs than the non-ε genes. However, this finding was not in agreement with our results; although the ε group genes contained more motifs, not all ε group genes contained more exons than non-ε group genes (Figure 1B,C). For instance, the ε group gene *TaGRF-like1* contained four exons, whereas the non-ε group gene *TaGRF2-B* contained six exons. This difference may be due to the insertion and deletion of introns over the long-term evolution of wheat. Nevertheless, gene structure and motif distribution were diverse between the two groups but similar within a group, which supports the results of the phylogenetic analysis (Figure 1B,C).

Common wheat is an allohexaploid with a large and complex genome, which makes its genome research difficult. Genome sequencing of common wheat and its sub-genome donors has been completed, which provides an important reference for the evolutionary analysis of wheat. The chromosome map of common wheat shows that the distance between genes is much greater than 200 kb (Figure 2A), which means that these genes were not generated by tandem duplication [43]. Collinearity analysis of common wheat and its ancestors revealed the direct source of the wheat 14-3-3 sub-genomic donor (Figure 2B), indicating that the entire family of wheat 14-3-3 proteins originated by polyploidization. The Ka/Ks ratio of 14-3-3 genes was <1 (Figure 2C), implying that wheat 14-3-3 proteins may remain unchanged during the process of long-term domestication. Analysis of *cis*-acting elements in the promoter region of *TaGRFs* indicated that these genes may be involved in biotic and abiotic stress responses, growth and development, and phytohormone response (Figure 3A). Previously, there have been some reports showing the involvement of 14-3-3 proteins in abiotic stress tolerance [44,45]. Considering the potential function of 14-3-3 proteins in abiotic stress, we performed transcriptome analysis of wheat plants under salt stress. Different *TaGRF* genes showed different expression profiles under salt stress. Most *TaGRF* genes were down-regulated after the salt treatment, suggesting that these genes may negatively regulate the response to salt stress (Figure 4A). However, in rice, tomato, *Brachypodium distachyon*, and other plant species, members of the 14-3-3 family are involved in salt stress tolerance [22,25,46]. In addition, wheat *TaGF14b* (named as *TaGRF1-B* in this article) also enhanced salt stress tolerance when ectopically expressed in tobacco [26]. In the current study, we selected two genes for the salt treatment and found that *TaGRF6-A* was changed greatly responding to salt stress (Figure 4B). To further verify the role of 14-3-3 family genes in salt stress, we used VIGS technology to silence the *TaGRF6-A* gene. The results showed that the level of leaf curling and number of BMSV: *TaGRF6-A* were more than the wild type (Figure 5B–D), indicating that *TaGRF6-A* plays a positive role in response to salt stress.

To elucidate the possible mechanism of action of *TaGRF6-A* under salt stress, we examined the expression of seven stress-related genes or potential interacting partners of *TaGRF6-A*. Interestingly, the expression level of *TaMYB64* was significantly down-regulated after *TaGRF6-A* silencing (Figure 6A). Since the interaction between MYBS2 and 14-3-3 protein has been reported in rice and soybean [32,33,47], we cloned the homolog of OsMYBS2 in wheat and performed Y2H, BiFC, and CoIP assays. The results showed that TaMYB64 and TaGRF6-A did interact in wheat (Figure 6C–E). Thus, we have the question of whether they use this interaction to improve salt tolerance. Next, we used the same method to silence *TaMYB64* in wheat, and the results proved our deduction. Compared with the control, the silenced plants also have a higher degree and amount of curling (Figure 7B,D). In summary, the results in the present study revealed the function of *TaGRF6-A* in salt stress, and initially elucidated its salt tolerance mechanism. Moreover, our study improved our understanding of the biological functions of the wheat *14-3-3* gene family.

## 4. Materials and Methods

### 4.1. Plant Material and Salt Treatment

Wheat (*T. aestivum* L.) cultivar Emai 170 was used in this study. Seeds of uniform size were selected and soaked in clear water for 24 h. Then, the seeds were sown in a small pot filled with nutrient-rich soil, and the pots were placed in a greenhouse maintained at 23 ± 2 °C, 16 h light/8 h dark photoperiod, and 200 lux light intensity. At the 2–3-leaf stage, wheat plants were treated with either double-distilled water (control) or 300 mM NaCl solution. Leaves were collected at 2, 4, 6, 12, 24, and 96 h after the treatment, frozen in liquid nitrogen and stored at −80 °C.

### 4.2. Mining and Phylogenetic Analysis of Wheat 14-3-3 Genes

To identify potential members of the wheat 14-3-3 gene family, published 14-3-3 amino acid sequences of *A. thaliana*, *O. sativa*, and *G. max* were used as queries in BLASTp searches against the wheat genome IWGSC RefSeq v.1.1, with an *E*-value cut-off <10^−10^. The HMM profile of the conserved domain of 14-3-3 (PF00244) was downloaded from the PFAM 32.0 database [48] and used to examine all wheat protein sequences with the HMMER search tool [49]. According to the sequence ID of the target candidate 14-3-3 protein, genomic sequences were isolated. Then, all matching sequences were validated using the SMART website (Simple Modular Architecture Research Tool, http://smart.embl-heidelberg.de/ (accessed on 13 February 2021)) and NCBI-CDD (National Center for Biotechnology Information-Conserved domain database, https://www.ncbi.nlm.nih.gov/Structure/bwrpsb/bwrpsb.cgi (accessed on 13 February 2021)) search to identify the conserved 14-3-3 protein domain with default parameters, and proteins lacking the 14-3-3 domain were discarded [50,51]. Previously known 14-3-3 protein sequences from *A. thaliana* (13), *O. sativa* (8), and *G. max* (18) used in this study were retrieved from the Phytozome v.12 database (http://phytozome.jgi.doe.gov (accessed on 13 February 2021)) [52].

### 4.3. Amino Acid Sequence Alignment and Characterization of TaGRF Proteins

Multiple sequence alignments were performed using DNAMAN software (version 7.212, Lynnon Corp., Quebec, QC, Canada). Secondary structures and three-dimensional models were constructed using Protein Homology/analogY Recognition Engine v.2.0 (Phyre2, http://www.sbg.bio.ic.ac.uk/phyre2/html/ (accessed on 13 February 2021)) [53]. The basic physicochemical properties of the 14-3-3 proteins, including MW, pI, and the number of amino acid residues, were examined using ProtParam on the ExPASy website (https://www.expasy.org/vg/index/protein (accessed on 13 February 2021)) [54].

### 4.4. Gene Structure Analysis and Conserved Motif Prediction

The GFF3 annotation file was obtained from the wheat reference genome IWGSC RefSeq v.1.1. Gene structure analysis was conducted using the Gene Structure Display Server 2.0 (GSDS, http://gsds.cbi.pku.edu.cn (accessed on 13 February 2021)) [55]. Conserved motifs in protein sequences were analyzed by MEME (http://meme-suite.org/tools/meme (accessed on 13 February 2021)) [56], with the maximum number of motifs set to 10.

### 4.5. Chromosome Distribution, Synteny, Ka/Ks, and Phylogenetic Analysis of TaGRFs

Information about the start and end of *TaGRFs* was extracted from the GFF3 file. A physical map of *TaGRFs* was constructed using MapInspect software Version 1.0 (http://www.softsea.com/review/MapInspect.html (accessed on 13 February 2021)) [57]. The TBtools software (https://github.com/CJChen/TBtools/ (accessed on 13 February 2021)) was used to determine the Ka and Ks values of *TaGRFs*, based on their coding sequences (CDSs) [58]. Reference genome sequences of wheat sub-genome donors were downloaded from NCBI, and *14-3-3* genes of each species were identified using the same methods as those used for determining the *TaGRFs*. To determine the paralogous or orthologous relationship between wheat *TaGRFs* and *14-3-3* genes of its sub-genome donors, the general tool “all against all BLAST searches” was used, with an *E*-value of 1 × 10^−10^ and sequence similarity > 75% [59]. The “circlize” package of the R program was used to draw the relationship between wheat *TaGRFs* and *14-3-3* genes of its sub-genome donors [60]. Phylogenetic analysis was conducted using the ML method of MEGA7, based on the aligned *14-3-3* sequences of *T. aestivum*, *A. thaliana*, *O. sativa*, *G. max*, *P. trichocarpa*, *M. truncatula*, and *V. vinifera*, with 1000 bootstrap replications [61]. The phylogenetic tree file was then uploaded to the Interactive Tree of Life (https://itol.embl.de/ (accessed on 13 February 2021)) for adjustment and modified [62].

### 4.6. Cis-Acting Element Analyses

The 1.5 kb genomic DNA sequences upstream of the start codon (ATG) of each *TaGRFs* genes were extracted from the wheat genome sequence. *Cis*-regulatory elements in the promoters were identified using the PlantCARE database (http://bioinformatics.psb.ugent.be/webtools/plantcare/html/ (accessed on 13 February 2021)) [63].

### 4.7. Expression Analysis of TaGRF Genes under Abiotic Stress Conditions

To determine the expression patterns of 14-3-3 genes in wheat under salt stress, wheat transcriptome data were downloaded from the NCBI Short Read Archive (SRA) database and mapped onto the reference genome of wheat using hisat2. The FPKM (fragments per kilobase of transcript per million) values obtained after “cufflinks” assembly were log-transformed, and a heatmap was drawn using the RStudio software “pheatmap” to display the expression profiles of *TaGRFs* [64,65].

### 4.8. RNA Isolation and qRT-PCR Analysis

Total RNA was extracted from leaf and root tissues of wheat plants treated with salt or water (control) using the RNAprep Pure Plant Kit (Invitrogen). The isolated total RNA was reverse transcribed to synthesize cDNA using the HiScript^®^ II 1st Strand cDNA Synthesis Kit (Vazyme) for qRT-PCR analysis. The cDNA was diluted to 100 ng/µL with RNase-free water, and qRT-PCR was performed in a 10 µL reaction volume containing 5 µL ChamQ Universal SYBR qPCR Master Mix, 0.5 µL each forward and reverse primer (10 µM), and 4 µL cDNA template. The following conditions were used for PCR: initial denaturation at 95 °C for 3 min, followed by 40 cycles of denaturation at 95 °C for 10 s, and annealing at 60 °C for 30 s. Fluorescence signals were collected after each cycle, and the temperature was increased from 60 °C to 95 °C after each cycle for melting curve analysis. The EF-1α gene (GeneBank accession: BT009129.1) was used as a reference gene. The relative expression level of genes was calculated using the 2^−∆∆CT^ method [66]. Three technical repeats were performed for each sample, and three independent replicates were carried out. Primers used for qRT-PCR are listed in Appendix A.

### 4.9. Virus-Induced Gene Silencing (VIGS) Assay in Wheat

There are four kinds of vectors involved in the VIGS test: α, β, γ, and γ-PDS. The gene fragments were cloned into the γ vector to obtain a recombinant vector. The vector (α, β, γ, γ-PDS, and recombinant vector) were linearized, and the linearized plasmid was treated with RiboMAX ™ Large Scale RNA Production System-T7 and the Ribom7G Cap Analog (Promega) to obtain capped in vitro transcription products. VIGS inoculation was carried out at the 3-leaf stage [67]. Steps are as follows: Mix equal volumes of in vitro transcription products α, β, γ (or γ-PDS /recombinant γ), dilute with DEPC water, add 1 × FES buffer (0.1 M glycine, 0.06 M K2HPO4, 1% *w*/*v* tetrasodium pyrophosphate, 1% *w*/*v* bentonite, and 1% *w*/*v* celite, pH 8.5), and then rubbed onto wheat leaves. BSMV: γ-PDS (PDS: wheat phytoene desaturase gene) and BSMV: γ were used as controls for BSMV infection. After inoculation, when BSMV: γ-PDS showed bleaching and yellowing phenomenon (about 10 days later), ddH_2_O and 300 mmol NaCl solution were used for irrigation. After 12 days of irrigation, the curl phenotype of the fourth leaf was recorded and the curl rate was counted. Curl rate is the percentage of curled leaves in all leaves.

### 4.10. Subcellular Localization of the TaGRF6-A Protein

The CDS of *TaGRF6-A* was cloned into the pQBV3 Gateway entry vector and then cloned into the pEarlyGate101 destination expression vector [68]. The resulting TaGRF6-A-YFP fusion construct was transformed into *Agrobacterium tumefaciens* strain GV3101, which was grown on LB (add antibiotics: kanamycin, rifampicin, gentamicin) solid medium for 2 days. The positive colonies were verified by PCR and transferred to LB (add antibiotics: kanamycin, rifampicin, gentamicin) liquid medium. The culture was grown for 16 h at 28 °C on a shaker until the optical density of the culture (measured at 600 nm absorbance; OD600) reached 1.5–1.8. The cells were harvested by centrifugation at 4000× g for 15 min and resuspended in acetosyringone (AS) culture solution. Then, 1 mL culture (OD600 = 0.8) was injected into the abaxial surface of the leaves of 3–4-week-old *N. benthamiana* plants using a needleless syringe, followed by incubation in the dark for 4 h. At 48 h post-inoculation, the distribution of the YFP signal in leaf epidermal cells was observed under a confocal laser scanning microscope (Zeiss LSM710) [69].

### 4.11. Y2H, BiFC, and CoIP Assays

The CDSs of *TaGRF6-A* and *TaMYB64* were cloned into pGBKT7 (BD) and pGADT7 (AD) vectors, respectively. According to the Yeast Protocols Handbook (Clontech, Mountain View, CA, USA), the recombinant plasmids were transformed into the yeast AH109 strain (*Saccharomyces cerevisiae*) and plated on an SD/-LW selection medium. The plates were incubated at 30 °C for 3–5 days until the appearance of colonies. Single colonies were picked using an inoculation ring and streaked onto SD/-LW and SD/-LWHA solid media. Plates were incubated at 30 °C for 3–5 days, and photographs were taken to record the growth of yeast colonies. T + P53 and T + lam served as positive and negative controls, respectively. Full-length cDNA sequences of *TaGRF6-A* and *TaMYB64* minus the stop codon were PCR amplified using the Pfu polymerase (NEB). The PCR products were ligated into the pQBV3 vector and then cloned into pEarleyGate201-YN and pEarleyGate202-YC vectors using the LR enzyme (Gateway LR Clonase II Enzyme mix, Invitrogen). The resulting plasmids were transformed into *A. tumefaciens* strain GV3101 and then transiently expressed in *N. benthamiana* leaves using the method described above. For Co-IP analysis, the PCR products were constructed on pEarleyGate100 and pEarleyGate104 vectors. Using the method of Qiao et al. [69], the protein was transiently expressed in *N. benthamiana* and the total protein was extracted, and then incubated with HA magnetic beads (MBL, Tokyo, Japan) at 4 °C and enriched with magnetic beads on ice. The precipitated protein was then separated by SDS-PAGE electrophoresis. Use anti-FLAG and anti HA antibody (MBL, Japan) to detect the coprecipitation signal of TaMYB64-YFP-3FLAG and TaGRF6-A-YFP-HA.

### 4.12. Statistical Analysis

Student’s *t*-test were applied to test differences among treatments.

## Figures and Tables

**Figure 1 ijms-22-01904-f001:**
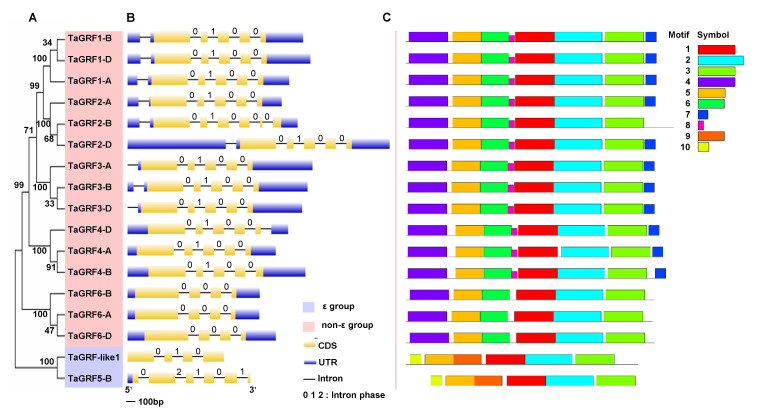
Structural analysis of *TaGRFs*. (**A**) The phylogenetic tree of *TaGRFs*. (**B**) Gene structure analysis of *TaGRFs* was conducted using the GSDS database. All introns are shown the same length. (**C**) Motif analysis of *TaGRFs* was identified by MEME. Motif components in the amino acid sequence of *TaGRFs* are displayed using MAST. Different motifs were represented in different color blocks.

**Figure 2 ijms-22-01904-f002:**
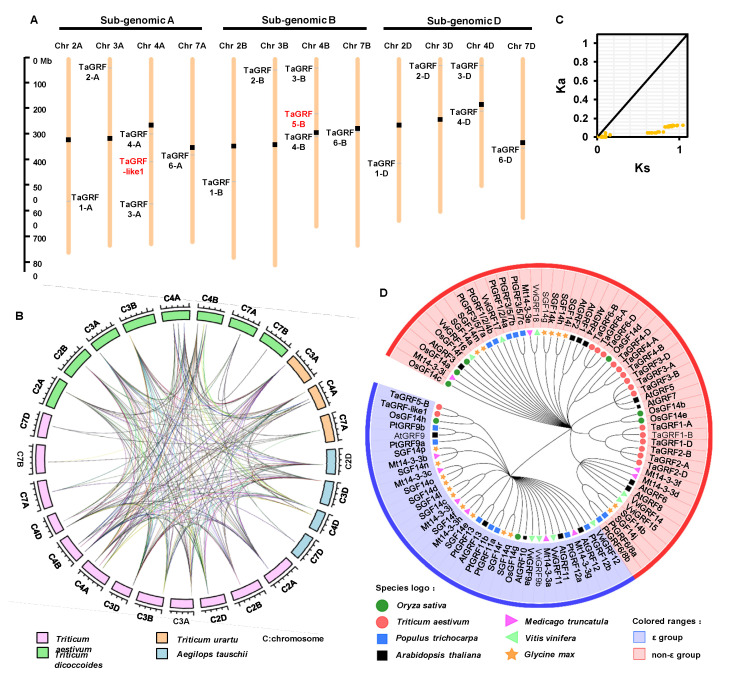
Evolutionary analysis of TaGRFs (**A**) Chromosomal locations and repeated events of TaGRFs. Orange bars represent chromosomes. The black rectangular dots above is the centromere. The scale indicates the corresponding physical distance. Chr 2A is the second chromosome on sub-genome A. The ε group genes are shown in red font, and non-ε group genes are shown in black font. The segmentally duplicated homologous genes were marked with blue lines. (**B**) Orthologous relationships analysis among T. aestivum (Ta, AABBDD, pink box) and progenitor species Ae. tauschii (Ae, DD, blue box), T. urartu (Tu, AA, orange box), and T. dicoccoides (Td, AABB, green box). The lines represent collinearity between chromosomes. (**C**) Ka/Ks value of TaGRF repeat gene pair. (**D**) Phylogenetic analysis of TaGRF proteins and other species such as *A. thaliana*, *O. sativa*, *G. max*, *P. trichocarpa*, *M. truncatula*, and *V. vinifera*. ClustalW2 was used to align the protein sequences, and the phylogenetic tree was constructed by the Neighbor-Joining (NJ) method of MEGA 7 with 1000 bootstrap replicates. Then the tree was modified by ITOL online tool. The two major groups were marked with different colors.

**Figure 3 ijms-22-01904-f003:**
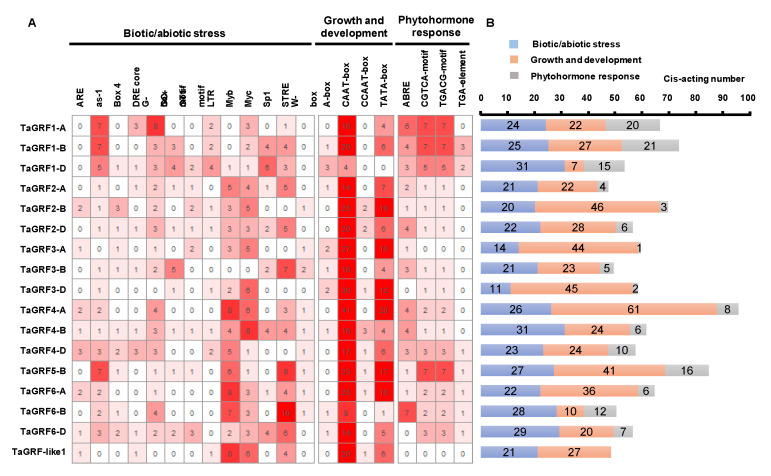
Analysis of *cis*-acting elements in the promoter of *TaGRFs*. (**A**) Different numbers and shades of color in the grid indicated the number of different promoter elements. (**B**) The histogram showed the sum of the cis-acting elements in different classes of each gene.

**Figure 4 ijms-22-01904-f004:**
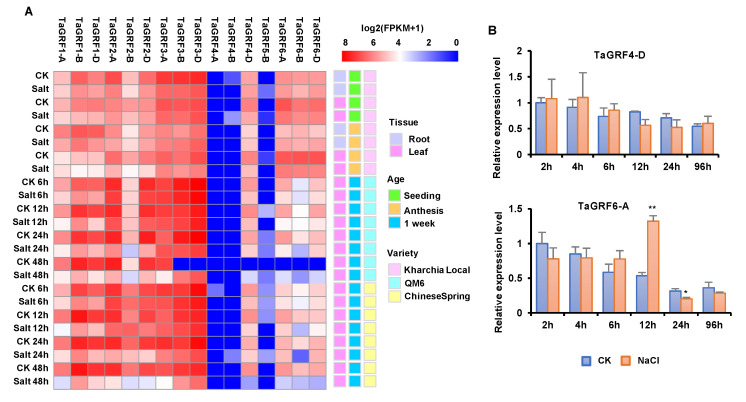
Expression patterns of *TaGRFs*. (**A**) Transcriptome analysis of *TaGRFs* across salt stress. The expression data were generated from NCBI and viewed in RStudio software. The relative expression level of a particular gene in each column was normalized against the mean value by log2 transformation. The color scale represents relative expression levels. (**B**) qRT-PCR analyses of two *TaGRF* genes in leaves under NaCl treatment. The asterisks indicate statistical significance between CK and NaCl treatments (**, *p* < 0.01; *, *p* < 0.05; Student’s *t*-test).

**Figure 5 ijms-22-01904-f005:**
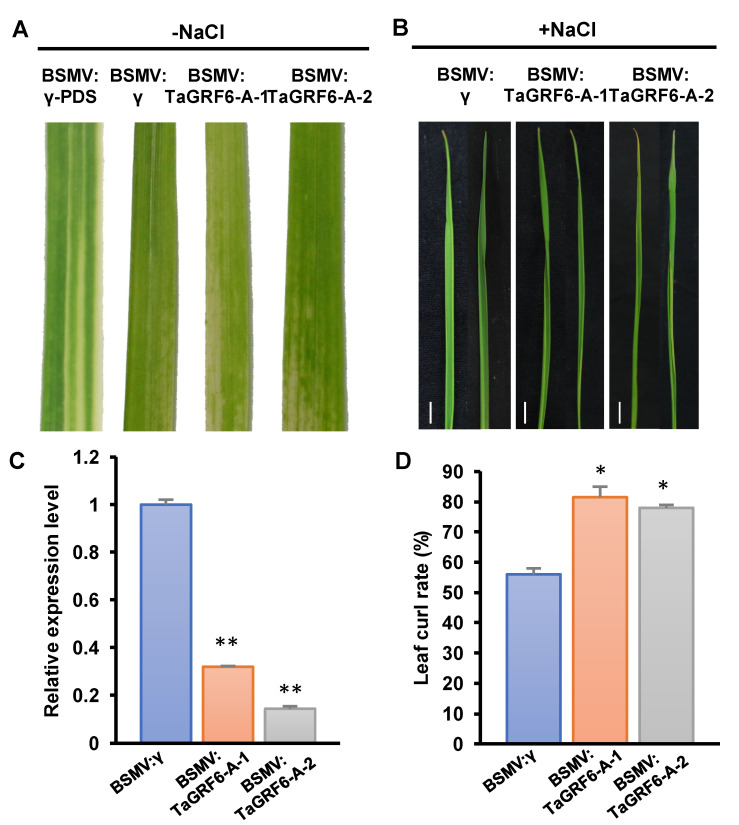
The silencing of *TaGRF6-A* decreases salt tolerance in wheat. (**A**) Phenotypic characteristics of wheat leaves after *TaGRF6-A* silenced by VIGS technology. (**B**) The curling phenotype of wheat leaves after 12 days of 300 mM NaCl treatment. Bars, 1 cm. (**C**) Silencing efficiency assessment of *TaGRF6-A* in the *TaGRF6-A*-knockdown plants treated with 12 days of 300 mM NaCl. (**D**) Statistics of leaf curls rate of *TaGRF6-A* in the *TaGRF6-A*-knockdown plants treated with 12 days of 300 mM NaCl. Error bars represent the SD of three biological replicates. The asterisks indicate statistical significance using Student’s *t*-tests. (**, *p* < 0.01; *, *p* < 0.05).

**Figure 6 ijms-22-01904-f006:**
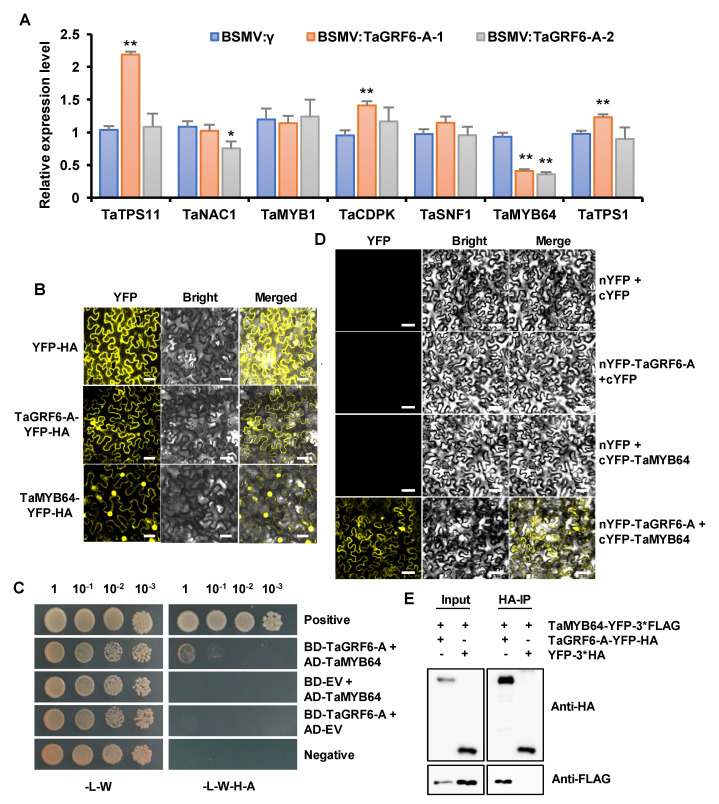
Analysis of protein interaction between TaGRF6-A and TaMYB64. (**A**) Relative expression levels of potential targets and stress-related genes in the leaves of *TaGRF6-A*-knockdown plants treated with 12 days of 300 mM NaCl. Error bars represent the SD of three biological replicates. The asterisks indicate statistical significance using Student’s *t*-tests. (**, *p* < 0.01; *, *p* < 0.05). (**B**) Subcellular localization of TaGRF6-A in *N. benthamiana*. Free YFP and TaGRF6-A-YFP and TaMYB64-YFP fusion proteins were transiently injected into *N. benthamiana* leaves by *Agrobacterium* (GV3101) transformation. Subcellular localization observed by laser confocal microscopy after 48 h post-infiltration. Scale bars, 40 µm. (**C**) Yeast two-hybrid interaction results of TaGRF6-A and TaMYB64 proteins. -L-W refers to media lacking leucine and tryptophan, -L-W-H-A refers to media lacking leucine, tryptophan, histidine, and adenine. (**D**) BiFC assay showing interactions of TaGRF6-A and TaMYB64 proteins. Scale bars, 40 µm. (**E**) Co-immunoprecipitation experiments show that TaGRF6-A interacts with TaMYB64 in planta. The total proteins of TaMYB64 with FLAG tag and TaGRF6-A with HA tag were extracted from *N. benthamiana* leaves. The immune complex was pulled down using anti-HA agarose gel, and the co-precipitation of TaGRF6-A and TaMYB64 was detected by Western blotting.

**Figure 7 ijms-22-01904-f007:**
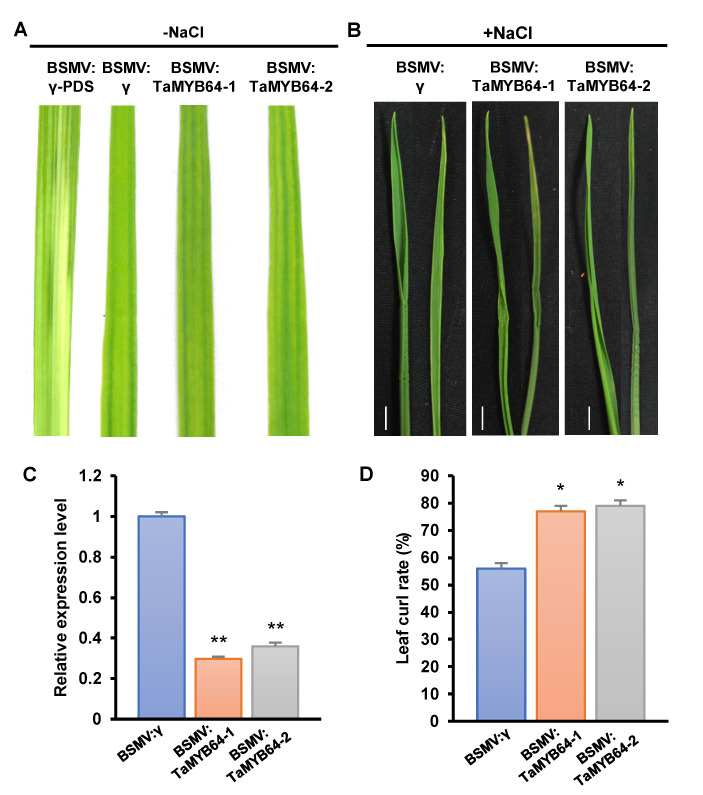
Silencing of *TaMYB64* decreases salt tolerance in wheat. (**A**) Phenotypic characteristics of wheat leaves after *TaMYB64* silenced by VIGS technology. (**B**) The curling phenotype of wheat leaves after 12 days of 300 mM NaCl treatment. Bars, 1 cm. (**C**) Silencing efficiency assessment of *TaMYB64* in the *TaMYB64*-knockdown plants treated with 12 days of 300 mM NaCl. (**D**) Statistics of leaf curls rate of *TaMYB64* in the *TaMYB64*-knockdown plants treated with 12 days of 300 mM NaCl. Error bars represent the SD of three biological replicates. The asterisks indicate statistical significance using Student’s *t*-tests. (**, *p* < 0.01; *, *p* < 0.05).

**Table 1 ijms-22-01904-t001:** Detailed information of putative TaGRFs.

Gene	Sequence ID	AS^a^	Genomic Position	CDS^b^	Protein^c^	MW^d^	pI^e^	Information
*TaGRF1-A*	TraesCS2A02G337300.1	2	Chr^f^2:570782629–57078596 (−)^g^	792	263	29.95	4.69	TaGF14a^#^
*TaGRF1-B*	TraesCS2B02G344600.1	2	Chr2:491180649–491184008 (+)^h^	792	263	29.97	4.73	TaGF14e^#^
*TaGRF1-D*	TraesCS2D02G325600.1	2	Chr2:419039525–419043028 (+)	792	263	29.95	4.69	TaGF14k^#^, TaGF14r^#^
*TaGRF2-A*	TraesCS3A02G055600.1	2	Chr3:32150045–32153332 (−)	789	262	29.69	4.67	TaGF14b^#^
*TaGRF2-B*	TraesCS3B02G068000.1	3	Chr3:40243799–40247161 (−)	849	282	31.93	4.98	Identified in this study
*TaGRF2-D*	TraesCS3D02G055500.1	1	Chr3:23061692–23065089 (−)	789	262	29.75	4.71	TaGF14m^#^, TaGF14s^#^
*TaGRF3-A*	TraesCS4A02G268700.1	2	Chr4:580905848–580908549 (−)	786	261	29.26	4.83	TaGF14d^#^
*TaGRF3-B*	TraesCS4B02G045500.2	2	Chr4:32805467–32808180 (+)	786	261	29.26	4.83	TaGF14h^#^
*TaGRF3-D*	TraesCS4D02G046400.2	2	Chr4:21886194–21888840 (−)	786	261	29.26	4.83	TaGF14j^#^
*TaGRF4-A*	TraesCS4A02G151000.2	3	Chr4:299599404–299602584 (−)	813	270	29.90	4.76	TaGF14c^#^
*TaGRF4-B*	TraesCS4B02G159900.1	3	Chr4:310973441–310976945 (+)	822	273	30.22	4.75	TaGF14g^#^
*TaGRF4-D*	TraesCS4D02G155900.1	2	Chr4:209095178–209098724 (+)	801	266	29.49	4.76	TaGF14n^#^
*TaGRF5-B*	TraesCS4B02G148900.1	2	Chr4:217846208–217849422 (+)	738	245	27.56	5.90	TaGF14i^#^
*TaGRF6-A*	TraesCS7A02G295500.1	1	Chr7:386346364–386357777 (−)	780	259	28.70	4.80	TaGF14o^#^
*TaGRF6-B*	TraesCS7B02G183800.1	1	Chr7:292828452–292841187 (+)	786	261	28.80	4.80	Identified in this study
*TaGRF6-D*	TraesCS7D02G292400.1	1	Chr7:356896682–356911158 (+)	786	261	28.80	4.74	TaGF14q^#^, TaGF14t^#^
*TaGRF-like1*	TraesCS4A02G167100.1	1	Chr4:412260859–412267803(+)	735	244	27.25	6.32	Identified in this study

AS^a^ (Number of alternative splices); CDS^b^ (Length of the coding sequence, bp); Protein^c^ (Amino acid length, aa); MW^d^ (Molecular weight, kD); pI^e^ (Isoelectric point); Chr^f^ (represents Chromosome); (−)^g^ (represents antisense strand); (+)^h^ (represents sense strand); reference^#^.

## Data Availability

Data is contained within the article or Appendix A.

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
