# Peer review of "Genome-Wide Identification and Characterization of Wheat 14-3-3 Genes Unravels the Role of TaGRF6-A in Salt Stress Tolerance by Binding MYB Transcription Factor"

_ijms, 2021, doi:10.3390/ijms22041904_

Round 1
Reviewer 1 Report
Comments to the Authors’
The manuscript entitled “Genome-Wide Identification and Characterization of Wheat 14-3-3 Genes Unravels the Role of TaGRF6-A in Salt Stress Tolerance by Binding MYB Transcription Factor” comprises the necessary elements of scientific novelty.
- Please see line 50 - italics Arabidopsis thaliana.
- Please see line 56-58 – italicize all the botanical names and also check throughout the manuscript.
- All the figures looking good. However, describe the more information in the results section.
- Please improve the discussion part with few recent references.
- Can you please differentiate and show the monocot and dicot plants in the Structural analysis of TaGRFs.
- The authors have to mention clearly for the readers, why did you choose some monocot and dicot species for comparison.
I would recommend the publication of this manuscript after addressing minor changes in the text and major revisions of the figure quality.
Author Response
Point 1: Please see line 56-58 – italicize all the botanical names and also check throughout the manuscript.
Response 1: Thank you for your kind reminding. I am deeply sorry for these avoidable mistakes. We have checked and corrected the full text.
Point 2: All the figures looking good. However, describe the more information in the results section.
Response 2: Thank you for your suggestion. We have checked and corrected.
Point 3: Please improve the discussion part with few recent references.
Response 3: Thank you for your suggestion. We have checked and corrected.
point 4: Can you please differentiate and show the monocot and dicot plants in the Structural analysis of TaGRFs.
Response 4: Because we are mainly studying the TaGRF gene family, the structural analysis only analysed in wheat. The following evolutionary analysis involves other species including monocot and dicot plant, because we want to predict the function of wheat genes based on the genetic relationship between different species and the known functions of adjacent genes.
Point 5: The authors have to mention clearly for the readers, why did you choose some monocot and dicot species for comparison.
Response 5: The 14-3-3 family proteins were well known conserved. The phylogenetic tree shows that the 14-3-3 family existed before the monocotyledonous and dicotyledonous plants diverged. Our main goal to construct this phylogenetic tree is to find some clues for the function of our TaGRF genes, which is adjacent to genes of known function in other species (either monocot or dicot). Usually the function of genes with closet relationship was similar.
Reviewer 2 Report
Paper "Genome-Wide Identification and Characterization of Wheat 14-3-3 Genes Unravels the Role of TaGRF6-A in Salt Stress Tolerance by Binding MYB Transcription Factor" is interestig but needs major revision.
My suggestions:
L30: "Triticum aestivum" not "triticum aestivum" and italic.
L34: "GENERAL REGULATORY FACTOR" - onlt first letters large.
L56-L58: Italic
L68: Italic
L69: Italic
L241: "2.5. Expression Profiling of 14-3-3 Genes" Lack of estimation of genes effects.
Figure 6A: Lack of LSD or HSD values.
The major minus of manuscript:
"Statisical analysis" section is ignored.
Paper needs major revision.
Author Response
Point 1: L30: "Triticum aestivum" not "triticum aestivum" and italic.
Response 1: Thanks for your suggestion. I am deeply sorry for these avoidable mistakes. We have checked and corrected.
Point 2: L34: "GENERAL REGULATORY FACTOR" - onlt first letters large.
Response 2: Thank you for your suggestion. I am deeply sorry for these avoidable mistakes. We have checked and corrected.
Point 3: L56-L58: Italic; L68: Italic; L69: Italic
Response 3: Thank you for your suggestion. I am deeply sorry for these avoidable mistakes. We have checked and corrected.
Point 4: L241: "2.5. Expression Profiling of 14-3-3 Genes" Lack of estimation of genes effects.
Response 4: Thank you for your kind reminding. We have made an assessment of gene effects based on gene expression. “the expression of TaGRF4-D did not change significantly, while the expression of TaGRF6-A was significantly up-regulated and then decreased. This trend suggests that TaGRF6-A can be induced by salt stress at a certain stage.”
Point 5: Figure 6A: Lack of LSD or HSD values.
Response 5: Thank you for your kind reminding. I think there may be some misunderstandings here. Although there are three columns in Fig.6A, it is not that three samples are compared in pairs, but both treatments are compared with the control. In this way, we only use the independent sample t test.
Point 6: The major minus of manuscript:"Statisical analysis" section is ignored.
Response 6: Thank you for your kind reminding. We have added a “Statistical analysis” section. “Statistical analysis: Student’s t-test were applied to test differences among treatments.”
Reviewer 3 Report
In this manuscript author did Genome-wide identification and characterization of wheat 14-3-3 genes unravels the role of TaGRF6-A in salt stress tolerance by binding MYB transcription factor.
Authors have identified total 17 potential 14-3-3 gene family members from the Chinese Spring whole-genome sequencing database.14-3-3 members in wheat exhibited significantly down-regulated expression in response to alkaline stress. VIGS assay and protein interaction analysis further testified that TaGRF6-A is positively regulated slat stress tolerance by interacting with a MYB transcription factor, TaMYB64. Manuscript is very well written and have enough result to be published at this point of time. I have few question regarding this manuscript.
In figure 5B why BSMV:Y showed only one leaf compared to the others.
Change at
L17 total of 17 to A total of 17.
L30 triticum aestivum to Triticum aestivum.
Use italic for all plant species.
I found plagiarism in some lines in this manuscript. L180-L185, L469-L473,
Author Response
Point 1: In figure 5B why BSMV:Y showed only one leaf compared to the others.
Response 1: Thanks for your kind reminding. Due to our negligence, only one picture was placed in the control group, which has now been added to the text.
Point 2: L17 total of 17 to A total of 17.
Response 2: Thank you for your suggestion. I am deeply sorry for these avoidable mistakes. We have checked and corrected.
Point 3: L30 triticum aestivum to Triticum aestivum.
Response 3: Thank you for your suggestion. I am deeply sorry for these avoidable mistakes. We have checked and corrected.
Point 4: Use italic for all plant species.
Response 4: Thank you for your kind reminding. I am deeply sorry for these avoidable mistakes. We have checked and corrected the full text.
Point 5: I found plagiarism in some lines in this manuscript. L180-L185, L469-L473.
Response 5: Thanks for your kind reminding. We are very sorry for the defect caused by our language negligence, and the language has been reorganized in the corresponding position.
L180-L185: “The lines represent collinearity between chromosomes. (C) Ka/Ks value of wheat TaGRF repeat gene pair. (D) Phylogenetic analysis of TaGRF proteins in wheat and other species such as A. thaliana, O. sativa, G. max, P. trichocarpa, M. truncatula, and V. vinifera. ClustalW2 was used to align the protein sequences, and the phylogenetic tree was constructed by the Neighbor-Joining (NJ) method of MEGA 7 with 1,000 bootstrap replicates. Then the tree was modified by ITOL online tool. The two major groups were marked with different colours.”
L469-L473: “The EF-1α gene (GeneBank accession: BT009129.1) was used as reference gene. The relative expression level of genes was calculated using the 2-∆∆CT method [66]. Three technical repeats were performed for each sample, and three independent replicates were carried out. Primers used for qRT-PCR are listed in Table S5.”

Round 2
Reviewer 2 Report
Thank you for all corrections. I recommend this manuscript to publication in present form.